# Small Peptide-Doxorubicin Co-Assembly for Synergistic Cancer Therapy

**DOI:** 10.3390/molecules25030484

**Published:** 2020-01-23

**Authors:** Shuangfei Li, Xianglan Chen, Huirong Chen, Jiaofeng Peng, Xuewei Yang

**Affiliations:** 1Shenzhen Key Laboratory of Marine Bioresource and Eco-Environmental Science, College of Life Sciences and Oceanography, Shenzhen University, Shenzhen 518060, China; szu_sfli@163.com (S.L.); cxlan12580@163.com (X.C.); chenhr@szu.edu.cn (H.C.); 2Instrument Analysis Center, Shenzhen University, Shenzhen 518060, China; jeffern@163.com

**Keywords:** peptide, Doxorubicin, co-assembly, delivery, cancer therapy

## Abstract

Design of elaborated nanomaterials to improve the therapeutic efficacy and mitigate the side effects of chemotherapeutic anticancer drugs, such as Doxorubicin (Dox), is significant for cancer treatment. Here, we describe a co-assembled strategy, where amphiphile short peptides are co-assembled with Doxorubicin to form nanoscale particles for enhanced delivery of Dox. Two kinds of short peptides, Fmoc-FK (FK) and Fmoc-FKK (FKK), are synthesized. Through adjusting the component ratio of peptide and Dox, we obtain two kinds of co-assembled nanoparticles with homogeneous size distributions. These nanoparticles show several distinct characteristics. First, they are pH-responsive as they are stable in alkaline and neutral conditions, however, de-assembly at acidic pH enables selective Dox release in malignant cancer cells. Second, the nanoparticles show an average size of 50–100 nm with positive charges, making them effective for uptake by tumor cells. Moreover, the side effects of Dox on healthy cells are mitigated due to decreased exposure of free-Dox to normal cells. To conclude, the co-assembled peptide-Dox nanoparticles exhibit increased cellular uptake compared to free-Dox, therefore causing significant cancer cell death. Further apoptosis and cell cycle analysis indicates that there is a synergistic effect between the peptide and Doxorubicin.

## 1. Introduction

Malignant tumors are one of the most lethal threats to human health. The major methods employed to treat cancer include chemotherapy, radiotherapy, and surgery. Concerning surgery and radiation therapies, they only can treat localized cancer. Regarding metastatic tumors, chemotherapy is the most prevalent treatment modality in the clinic. However, chemotherapeutics usually display severe side effects, thus, resolutions that mitigate the side effects while retaining the therapy effects of chemotherapeutics is still unmet and urgently needed. Doxorubicin (Dox), also known as Adriamycin (ADR), is a potent chemotherapeutic drug applied in the clinic for the treatment of a wide range of human cancers, including acute leukemia, lymphoma, breast cancer, lung cancer, bladder cancer etcetera [1]. Dox exerts its therapeutic effect mainly by two mechanisms [2,3]. It can insert into the base pairs of the DNA strands thus inhibiting DNA replication and RNA synthesis in cells. Moreover, Dox is a good electron acceptor in redox reactions. It can be oxidized into a semiquinone free radical, which causes oxidative damage to cellular membranes, proteins, and DNA by producing reactive oxygen species (ROS). However, Dox also exhibits severe side effects such as cardiac damage and bone marrow suppression, which largely limit the clinical application of Dox [4,5,6]. Encapsulation of Dox with liposomes is an effective strategy to change in vivo distribution, increase its anti-tumor effect, reduce its cardiac toxicity, and allow it to become a welcome product on the market [7,8,9]. However, liposomes show some limitations, including poor stability, drug leakage, short residence time, and inadequate dispersion [7,10,11,12,13]. Additionally, since Dox is difficult to dissolve under neutral and alkaline conditions, and only dissolves well under acidic conditions, it can convert to a hydrophilic form and partially dissolve into water [14,15]. Thus, the poor water solubility at neutral and alkaline conditions and the great toxic side effects are the key factors hindering its clinical application. 

Nanomaterials have demonstrated great potential in delivering and transporting drugs, permeating cell membranes and releasing drugs in tumor cells through the enhanced permeability and retention (EPR) effect. The advantages of nanomaterials include augmenting the accumulation of drugs in tumor cells while reducing the unwanted uptake by normal cells [16,17,18]. Driven by the need to reduce the side effects of Dox and enhance the specific cytotoxicity of Dox against cancer cells, numerous nanomaterials for controlled release of Dox were designed over the past several decades, including mesoporous silica, gold nanoparticles, and polymer micelles, etcetera [9,19,20,21,22,23,24,25,26,27,28,29,30,31]. Accompanying these strategies, the water solubility of Dox is improved and the high toxicity of Doxorubicin is mitigated. Despite great success, several limitations of the current nanoplatform still remain. Considering liposomes, for example, Dox leakage and inefficient cellular uptake usually occur [32]. Regarding other inorganic and polymeric nanoparticles, the long retention toxicity in the human body due to limited degradability also impede widespread application [33,34]. Basically, more favorable Dox delivery nanoplatforms are required.

An ideal drug-delivery nano-platform could present the following characteristics. It could maintain the drug in the nanoform under normal physical conditions while releasing the drug at tumor sites [35,36,37]. Thus, stimuli-responsive drug delivery platforms have been exploited aiming at the controllable release of anti-cancer drugs according to the difference between normal tissue and tumor sites. Among the most popular stimuli-responsive candidates for drug delivery systems, pH-responsiveness is one of the most prevalent methods owing to the nuanced distinction between acidic cancer cells (pH 6.5–5.0) and normal physical conditions (pH 7.4) [38,39,40,41]. Peptide-based supramolecular nanomaterials have been extensively investigated as drug delivery vehicles due to their excellent properties of biocompatibility, easy accessibility, and tunable functionalization [42,43,44]. Two types of strategies are usually used in this area. The first type is peptide nanomaterials as a delivery vehicle and chemotherapeutic drugs as payloads [45,46,47]. Here, the chemotherapeutic drug is not an assembly component of the nanomaterials. Regarding the second, a chemotherapy drug is a necessary component to drive the successful assembly. Compared to the first type, the second type shows advantages on the controllable release of Dox [48,49].

We expect that Dox can be used to construct a co-assembly with amphiphile peptides due to its aromatic chemical attributes. Based on this hypothesis, we design and synthesize two short peptides, FK and FKK, and their co-assembly behavior with Dox are systematically investigated. Two peptide-Dox co-assembled nanoparticles are selected for bioactivity evaluation (Figure 1). We find that both of the co-assemblies show a more effective cellular uptake and cancer cell killing effects. This phenomenon suggests that potential synergistic effects exist between peptides and Dox for promoting cellular uptake and inducing cell apoptosis and cell cycle arrest.

## 2. Results and Discussion

### 2.1. Preparation and Characterization of Nanoparticles 

Figure 2A shows the chemical structures and the representative three-dimensional conformation of Doxorubicin (Dox), Fmoc-FK (FK), and Fmoc-FKK (FKK). Viewing these structures, we can see that all of these compounds possess aromatic functional groups. Specifically, the Dox possesses three conjunctive aromatic rings, and the Fmoc-capping group in Fmoc-FK and Fmoc-FKK is a well-known gelation inducer which has been reported to induce the formation of nanofibers of hydrogels. Based on these, we hypothesized that Dox could form co-assemblies with Fmoc-FK or Fmoc-FKK via aromatic interaction. Figure 2B shows a proposed model of the molecular packing between Dox and FK/FKK. Using this hypothesis, we then tried to exploit the assembly behavior of these molecules.

To exploit the most suitable conditions for assembly, we prepared a series of peptide and Dox mixtures with different component ratios. To access the quality of assembled materials, we measured the size distribution and polydispersity index (PDI) of these materials by dynamic light scattering (DLS). Consequently, two components, (FK)_2_-Dox and FKK-Dox, showed favorable diameters and PDI. Regarding (FK)_2_-Dox, it means that the molar ratio of FK peptide against Dox was 2:1. The average diameter was 50–100 nm, with a PDI of less than 0.2, indicating a homogeneous assembly. Concerning FKK-Dox, the molar ratio of the average diameter was slightly larger than that of (FK)_2_-Dox, showing between 80–150 nm, with a PDI of less than 0.3. These size distributions were located at a suitable size range for EPR effects. Regarding other components, the size distributions of the assembled nanomaterials were too large, as well as having a large PDI, suggesting that the assembly materials displayed uneven sizes. 

We then, with two assembled nanomaterials in hand, checked their morphology by using scanning electron microscopy (SEM). Figure 3B suggests both (FK)_2_-Dox and FKK-Dox form nanoparticles and the diameter of the nanoparticles is 50–200 nm, consistent with that observed using DLS. Furthermore, we employed DLS to measure the zeta potential of these nanoparticles (Figure 3C). Both of the two kinds of nanoparticles were positively charged, and the FKK-Dox showed a zeta potential of 18 mV, which was a little higher than that of (FK)_2_-Dox (15 mV). The positive zeta potential suggested that the nanoparticles were favorable for cell membrane attachment and endocytosis.

### 2.2. In Vitro pH-Dependent Dox Release

One of the features of the co-assembled nanoparticles was that the Dox release occurred when the nanoparticles were disassembled. This kind of drug release profile prevents unwanted Dox leakage or uncontrollable release incidence when using liposomes as carriers. To evaluate the disassembly kinetics of the nanoparticles at different pH, we measured the Dox release profiles. Shown in Figure 4, at pH 8.5 and 7.0, the release rates of Dox in both of the nanoparticles were very slow; nearly 15 wt% of the encapsulated Dox was released at 16 h. When the pH value of the incubation solution was changed to 5.0, the release rate of Dox was significantly increased, as 55 wt% and 65 wt% of Dox was released from (FK)_2_-Dox and FKK-Dox, respectively. This Dox release behavior indicated that our nanoparticles were pH-responsive, enabling the precise delivery and release of Dox in the tumor’s acidic microenvironment. 

### 2.3. Cellular Uptake of the Nanoparticles

Cell uptake and intracellular Dox release of the nanoparticles were evaluated with flow cytometry (Figure 5) and confocal laser scanning microscopy (CLSM) (Figure 6), respectively. Shown in Figure 5A, both the (FK)_2_-Dox and FKK-Dox showed a higher uptake in Hela cells than that of Dox. This phenomenon is mostly caused by the positive charge of nanoparticles facilitating the endocytosis of the nanoparticles. Conversely, the FKK-Dox nanoparticles exhibited a higher uptake than (FK)_2_-Dox (Figure 5B), consistent with the larger positive zeta potential of FKK-Dox. Figure 6A shows the live-cell confocal images of free-Dox and the nanoparticles. Similar to free-Dox, most of the fluorescence in the nanoparticle-treated groups was emitted from the cell nucleus, indicating the Dox was successfully released from the nanoparticles into the cells. The quantitative analysis of the cellular fluorescent intensity showed an order of the intensity of FKK-Dox > (FK)_2_-Dox > Dox, in good agreement with the fluorescence-activated cell sorting (FACS) results. 

### 2.4. In Vitro Cytotoxicity of the Nanoparticles

The cytotoxicity of nanoparticles was determined using a CCK-8 assay in MDA-231, A549, and Hela cells. Cells were treated for 48 h at various Dox concentrations ranging from 0 to 1.6 µM mL^−1^. Shown in Figure 7, the cell viability decreased obviously as the nanoparticle concentration increased. We estimated the half maximal inhibitory concentration (IC_50_) of the free-Dox, (FK)_2_-Dox, and FKK-Dox in Figure 7D. Among all the cell lines, the FKK-Dox showed the lowest IC_50_, followed by (FK)_2_-Dox, and the free-Dox showed the largest IC_50_. Specifically, the IC_50_ of FKK-Dox in MDA-231 was 167.9 ± 35.99 nM, over two folds lower than that of free-Dox (422 ± 66.76 nM). The IC_50_ was the lowest in Hela cells. The different IC_50_ was consistent with the cell uptake and intracellular Dox release data. Figure 7E is the optical view of Hela cells after treatment. The cell density of Hela cells after FKK-Dox treatment was lowest, and the cells displayed a round morphology, indicating a destroyed cell growth state. This phenomenon also was observed in Dox- and (FK)_2_-Dox-treated cells. Particularly, we also studied the cytotoxicity of FK and FKK; both of the peptides showed little toxicity to cell growth, even at high concentrations, as shown in Appendix A. These results showed the application potential of these nanoparticles in cancer therapy.

### 2.5. Cell Cycle Analysis and Cell Apoptosis

The Dox can insert into the grooves of DNA, thus inhibiting the normal mitosis of cancer cells. We employed flow cytometry to analyze the cell cycle distribution changes after nanoparticle treatment. Figure 8A–D is the cell cycle distribution results. Obviously, FKK-Dox induced the most G2/M phase arrest, with 32.36% of cells compared to 17.72%, 23.8%, and 25.16% for control, Dox, and (FK)_2_-Dox, respectively (Figure 8E). Then, an Annexin V-FITC Apoptosis assay was conducted to reveal whether the nanoparticles on Hela cells might result in enhancement of cell apoptosis. Shown in Figure 9, the rate of apoptosis of Hela cells showed an increasing trend. FKK-Dox nanoparticles led to the highest apoptosis (~60%), and it was about two-folds higher than that of free-Dox (~30%). The (FK)_2_-Dox resulted in ~42% cell apoptosis, located between the Dox and FKK-Dox. These results were consistent with cell viability and cellular uptake. Moreover, the reactive oxygen species (ROS) generation in the cells after being treated with Dox, (FK)_2_-Dox and FKK-Dox was monitored. Shown in Appendix A, the FKK-Dox induced the highest level of cellular ROS in Hela cells. These results further confirmed the apoptotic results.

## 3. Materials and Methods

### 3.1. Materials and Characterization

All peptide materials were purchased from Synpeptide Co., Ltd. (Nanjing, China) and Doxorubicin was obtained gratis from Zhejiang Hisun Pharmaceutical Co., Ltd. (Taizhou, China). Dulbecco’s modified Eagle’s medium (DMEM) was purchased from M and C Gene Technology Inc. (Beijing, China). Trypsin and ethylenediaminetetraacetic acid (EDTA) were purchased from Amresco (Solon, OH, USA). Fetal bovine serum (FBS) was purchased from Zhejiang Tianhang Biological Technology Co., Ltd. (Zhejiang, China). MDA-231, A549, and Hela cell lines were purchased from the Chinese Academic of Science (Shanghai, China). The SEM images were characterized by Hitachi S-3400N(II) (Japan) in the public experimental platform of Shenzhen University (Shenzhen, China). 

### 3.2. Preparation of Peptide-Dox Co-Assemblies 

The nanoparticles were prepared by an “organic solvent-water exchange method”. Briefly, the peptide (10 mM) and Dox (5 mM) were dissolved in hexafluoroisopropanol with designated molar ratios and stirred overnight at room temperature. Then, the mixture solution was dropped into water, and the solution was bubbled with N_2_ to remove the hexafluoroisopropanol. The resulted nanoparticles were further stirred at room temperature (r.t.) for two days to make homogeneous nanoparticles.

### 3.3. Dynamic Light Scattering and Zeta Potential Measurements

The diameter and particle size distribution of nanoparticles, such as Z-average diameter (Zavd), Polydispersity index (PDI), were measured by photon correlation spectroscopy (PCS) on a Malvern Zetasizer Nano ZS (Malvern Instruments, WR14 1XZ, UK). The surface charge was estimated by measuring the zeta potential (ZP) based on electrophoretic mobility without dilution.

### 3.4. In Vitro Dox Release

Nanoparticle (0.01 M of Dox) dispersed in phosphate buffered saline (PBS) at pH 8.5, pH 7.0 and pH 5.0 were transferred into a dialysis bag (Mw cut-off: 3500 Da). The bag was placed into the same buffered solution (150 mL), and the release study was performed at 37 °C in an incubator shaker (TS-100C, Shanghai Kuangbei, China). Three milliliters of the solution outside the dialysis bag was replaced with the same volume of fresh buffer solution for UV-vis analysis at certain time intervals. Dox concentration was calculated based on the absorbance intensity of Dox at 497 nm. Regarding the assessment of drug release, the cumulative amount of released drug was calculated, and the percentages of drug released from micelles were plotted against time.

### 3.5. Cell Culture

Human cervical carcinoma (HeLa) cells, triple-negative breast cancer MAD-231 and adenocarcinomas human alveolar basal epithelial cells were maintained and grown in Dulbecco’s modified Eagle’s minimum essential medium (DMEM) (Corning, Thermo Fisher Scientific, Waltham, MA, USA), supplemented with 10% fetal bovine serum (FBS) (Atlanta Biologicals, Lawrenceville, GA, USA) and 1% Penicillin/Streptomycin antibiotics (HyClone, Thermo Fisher Scientific). All cell lines were bought from the Chinese Academy of Sciences Shanghai Cell Bank (Shanghai, China) (http://www.cobioer.com/products_list/pmcId=55.html?b_scene_zt=1&renqun_youhua=1981537&bd_vid=8143897183949310144).

### 3.6. Fluorescence Microscopy 

Hela was seeded in 35-mm dishes for live-cell imaging in a Dulbecco’s Modified Eagle Medium (DMEM) medium with 10% fetal bovine serum and grown in a 5% CO2 incubator at 37 °C for 24 h. After removing the old medium, the cells were treated with free-Dox, peptide-Dox with final Dox (1 μM) in a fresh medium for 6 h. Then, the culture media was removed, cells were washed twice with cold PBS, Dox uptake (EX 480 nm, EM 590 nm) was observed and imaged with a Nikon confocal microscope (Nikon C2+, Nikon Instruments Inc., Melville, NY, USA).

### 3.7. Cellular Uptake Assay

Hela cells (1.0 × 105 cells/well) were plated in 6-well tissue culture plates and incubated for 24 h at 37 °C. After overnight incubation, cells were exposed to a range of concentrations of free-Dox or peptide-Dox for 24 h. To harvest the cells after the treatment, we used a non-enzymatic cell dissociation buffer, which did not degrade the polypeptide attached to the cells. Forward versus side scatter gating was used to remove cell debris from the analysis and Dox fluorescence intensity (*n* = 10,000 cells) was measured using a BD Bioscience FacsCanto II Flow Cytometer and Flowjo software (Becton Dickinson, San Jose, CA, USA). Fluorescence intensity was normalized to cellular auto-fluorescence.

### 3.8. Cell Viability Measurements

Hela cells were seeded with 1.0 × 104 cells in a 96-well plate and cultured at 37 °C and 5% CO_2_ atmosphere for 24 h. An equal volume mixture of DMEM (100 µL) and 6 h-matured-solution containing self-assembled peptide-Dox nanoparticles in a 20 mM phosphate buffer, 150 mM NaCl, pH 7.2 (100 µL) was prepared and subsequently transferred into the 96-well plate, and incubated at 37 °C and 5% CO_2_ atmosphere for 24 h. Cells were washed with 1× HEPES twice, followed by treatment with 1× HEPES (50 µL) and 0.2% Cell Counting-kit 8 (50 µL, Dojindo Molecular Technologies, Kumamoto, Japan) at 37 °C and 5% CO_2_ atmosphere for 30 min for cell viability measurements.

### 3.9. Apoptosis Assay

Apoptosis was measured by flow cytometry using BD Bioscience FacsCanto II Flow Cytometer (USA). Briefly, cells were seeded in transparent 6-well plates with a density of 1 × 10^5^ cells per well for Hela cells. After overnight incubation, cells were treated with either free-Doxorubicin or peptide-Dox at designated concentrations. After 24 h, both floating and attached cells were harvested and stained with conjugated fluorescein isothiocyanate (FITC) Annexin-V and Propidium iodide (PI). Forward versus side scatter gating was used to eliminate cell debris from the analysis, and a scatter plot of PI intensity versus FITC Annexin V intensity was used to score live, apoptotic, and necrotic cells. The percentage of Annexin-V positive, apoptotic cells was expressed as an average of three experiments.

### 3.10. Cell Cycle Analysis

Hela cells were plated in 6-well plates at a density of 1 × 10^5^. The following day, cells were treated with peptide-Dox or free-Doxorubicin for 24 h at 37 °C. After the treatment, the cells were rinsed with PBS, fixed with 3 mL ice-cold 70% ethanol for 30 min, rinsed with PBS again and resuspended in 500 µL PBS. To eliminate the signal from ribonucleic acid (RNA), we added RNase A (Sigma Aldrich, Saint Louis, MO, USA) to a final concentration of 750 µg/mL to the cell suspension for 5 min. Cells were then treated with 200 µg/mL of Propidium iodide (PI) (Sigma, St., Louis, MO, USA) for 30 min at room temperature. To evaluate the intensity of PI fluorescence, as a measure of DNA content, we used the BD flow cytometer and analyzed the results with Flowjo software. A plot of forward scatter versus PI intensity was gated to remove cell debris and cell aggregates from the analysis. Fluorescence was measured for a sample of 10,000 cells using FL2 (laser ex. 488 nm, filter 620/30 nm), and histograms of cell number versus PI intensity were used to determine the percentage of cells in each phase of the cell cycle.

### 3.11. Statistical Analysis 

All data were tested at least three times independently. Each experiment in the cell death assay was performed by 3–6 replicates. All data were represented as means ± standard deviation (SD) (*n* = 3). Student′s t-test was performed to assess the statistical significance. A *p*-value < 0.05 was considered statistically significant.

## 4. Conclusions

As a highly efficient chemotherapy drug, the liposomal Dox (Doxil) has been widely used to treat AIDS-related Kaposi′s sarcoma, breast cancer, ovarian cancer, and other solid tumors [50]. The success of Doxil relies on liposome formulation which significantly decreases the cardiotoxicity of free Dox. However, the liposomal Dox also shows some severe side effects, such as Hand-Foot Syndrome [51]. Considering this, the development of other forms of Dox is anticipated. Here, we demonstrated that elaborated short amphiphile peptides and Dox co-assembles could form nanoparticles which act as a new form of Dox delivery agent. Compared to free-Dox, the peptide-Dox shows several advantages. First, the cellular uptake of Dox is enhanced due to the positive charges of nanoparticles. Second, the release of Dox is pH-responsive; the co-assembled nanoparticles can be triggered to release the Dox due to the acidic pH in the tumoral microenvironment. Moreover, this pH-responsive drug release behavior is anticipated to mitigate the side effects of Dox to healthy tissue. Last but not least, the homogeneous and sub-100 nm size of the co-assembled nanoparticles makes them suitable for tumor lesion accumulation due to the EPR effects. To summarize, the peptide-Dox co-assembled nanoparticles are biodegradable, highly efficient, and pH-responsive. Next, we will test the efficacy of these nanoparticles in vivo to treat cancers. 

## Figures and Tables

**Figure 1 molecules-25-00484-f001:**
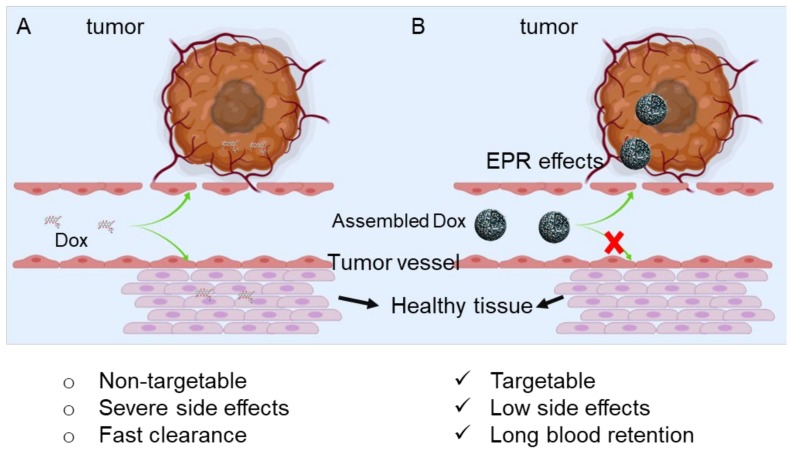
Schematic presentation of the delivery of Doxorubicin to tumor lesions via assembled nanoparticles (**B**) compared with non-specific administration of Dox (**A**). When the Dox is assembled into nanoparticles, the tumor targetability of the Dox can be increased by leveraging the EPR effects. Simultaneously, the side effects will be decreased. Also, the nanoformulation Dox will show a longer blood circulation time than free-Dox.

**Figure 2 molecules-25-00484-f002:**
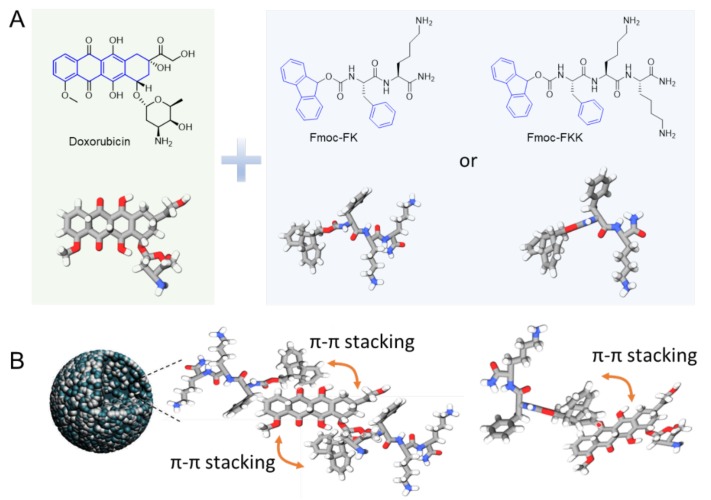
(**A**) Chemical structures and representative three-dimensional branch view of the Dox, Fmoc-FK, and Fmoc-FKK. (**B**) Proposed molecular packing in the assembled nanoparticles. It is anticipated that the π–π stacking interaction between aromatic rings will dominate the assembly kinetics.

**Figure 3 molecules-25-00484-f003:**
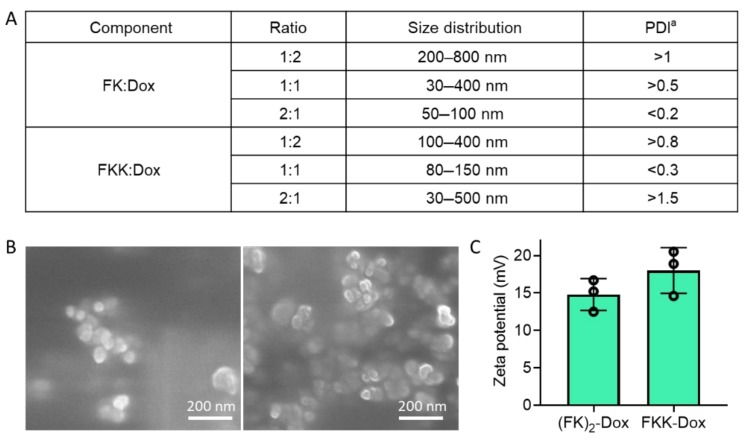
(**A**) Summary of the component ratio used in this study and the resulting size distribution and polydispersity index (PDI) of different components. ^a^ The value represents the mean of three times of replication. (**B**) SEM images of (FK)_2_-Dox (left) and FKK-Dox (right). (**C**) Zeta potentials of (FK)_2_-Dox and FKK-Dox.

**Figure 4 molecules-25-00484-f004:**
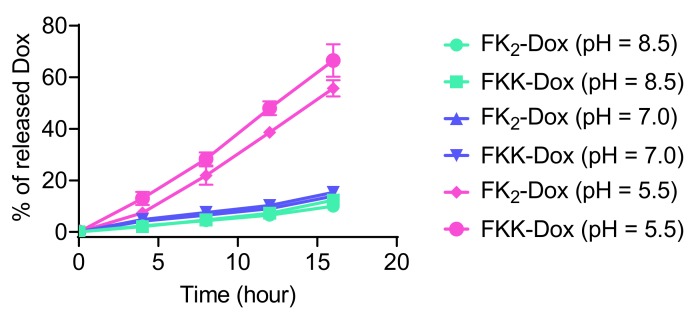
The in vitro Dox release kinetics from (FK)_2_-Dox and FKK-Dox assembled nanoparticles at pH = 8.5, 7.0, or 5.5. The values were obtained from three independent experiments. Error bars are standard deviation (SD).

**Figure 5 molecules-25-00484-f005:**
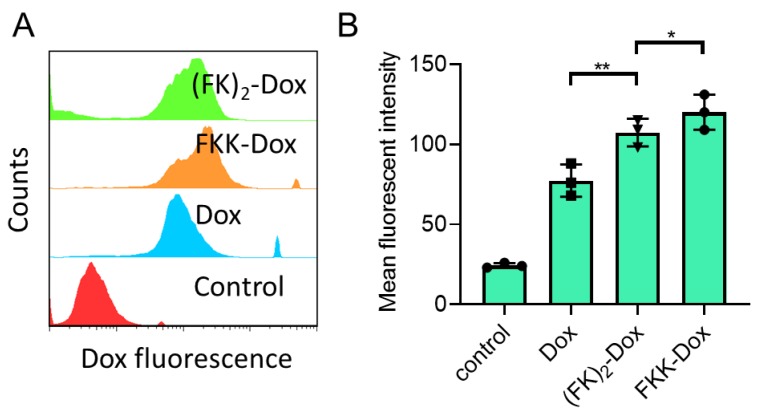
(**A**) Representative cellular uptake histograms obtained from cytometry analysis. Hela cells were treated with DMSO, Dox (1 µM), (FK)_2_-Dox (1 µM of Dox) and FKK-Dox (1 µM of Dox) for 4 h at 3 °C. (**B**) Quantitative analysis of cellular fluorescence of a Hela cell treated by different Dox species corresponding to Figure 5A. The values represent the average geometric mean of uptake for three independent experiments. *P* values were determined by the Student’s t-test. ** *P* < 0.01; * *P* < 0.05. Error bars are SD.

**Figure 6 molecules-25-00484-f006:**
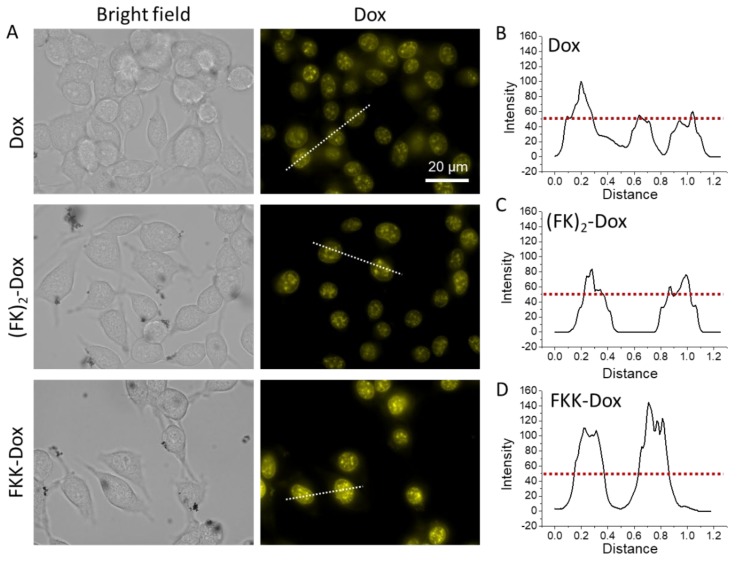
(**A**) Hela cell uptake of Dox (5 µM), (FK)_2_-Dox (5 µM of Dox) and FKK-Dox (5 µM of Dox) analyzed by fluorescence microscopy. The images were acquired with live cells. The cells were incubated with different Dox species for 6 h, after that, the medium was removed and replaced with fresh medium. The emission of Dox was designated with yellow color. (**B**–**D**) The fluorescent intensity alongside the white dashed lines in Figure 4A.

**Figure 7 molecules-25-00484-f007:**
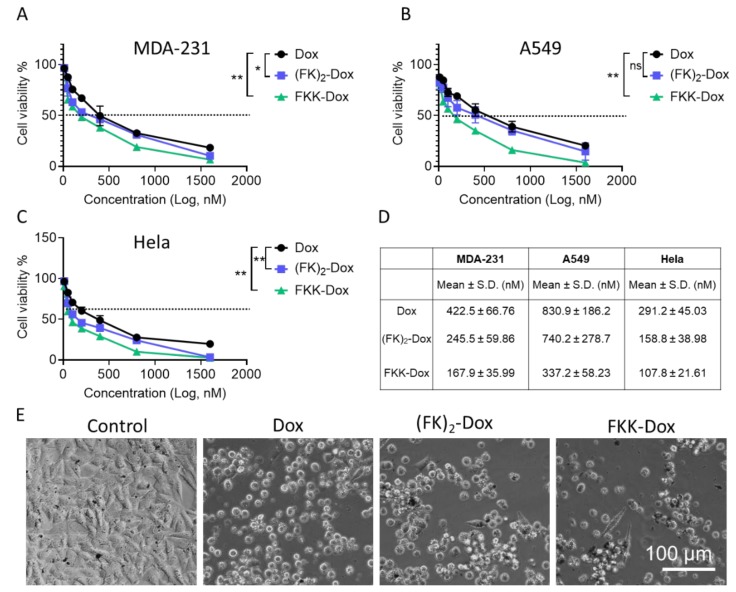
(**A**–**C**) Cell viability measurements for MDA-231, A549, and Hela cells treated with different concentrations of Dox, (FK)_2_-Dox or FKK-Dox. The values were measured at 2 days post-treatment of the Dox species. Three independent replications were performed. *P* values were determined by a repeated measures (RM) two-way analysis of variance (ANOVA) with Tukey’s multiple-comparisons test. ** *P* < 0.01; * *P* < 0.05. (**D**) Calculated IC_50_ values of different Dox species to inhibit the cancer cell growth. (**E**) The optical view of a Hela cell morphology after treated with Dox (300 nM), (FK)_2_-Dox (300 nM of Dox), and FKK-Dox (300 nM of Dox) for 24 h; the cells without any treatment were used as control.

**Figure 8 molecules-25-00484-f008:**
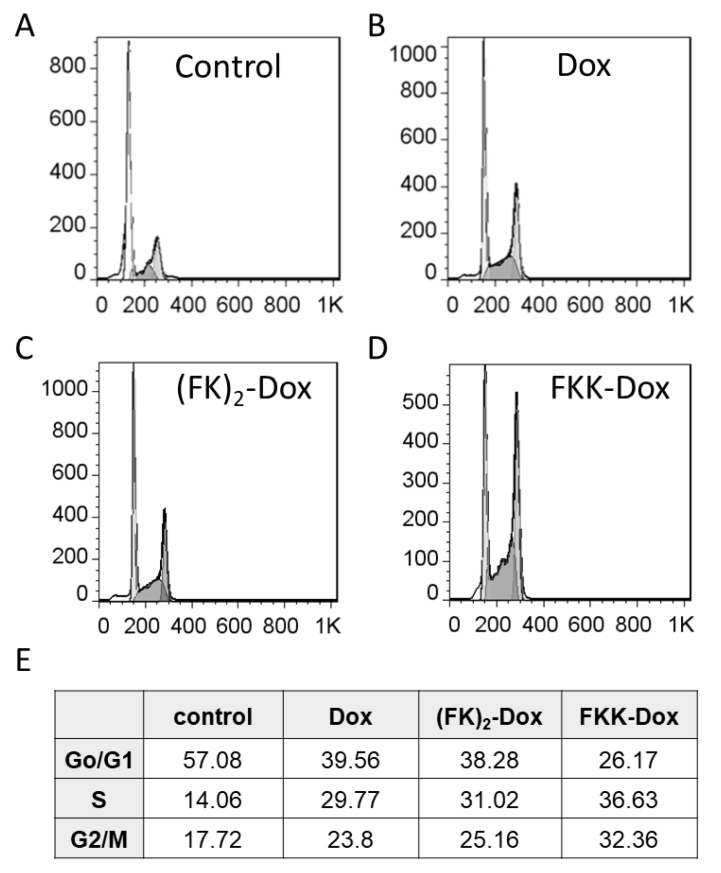
(**A**–**D**) Cell cycle distribution analysis. Hela cells were treated with free Dox (100 nM), (FK)_2_-Dox (100 nM of Dox) or FKK-Dox (100 nM of Dox) and analyzed at 24 h after the treatment. Raw data for a representative experiment is shown. (**B**) Quantitative analysis of cell cycle distribution based on Figure 8**A**–**D**.

**Figure 9 molecules-25-00484-f009:**
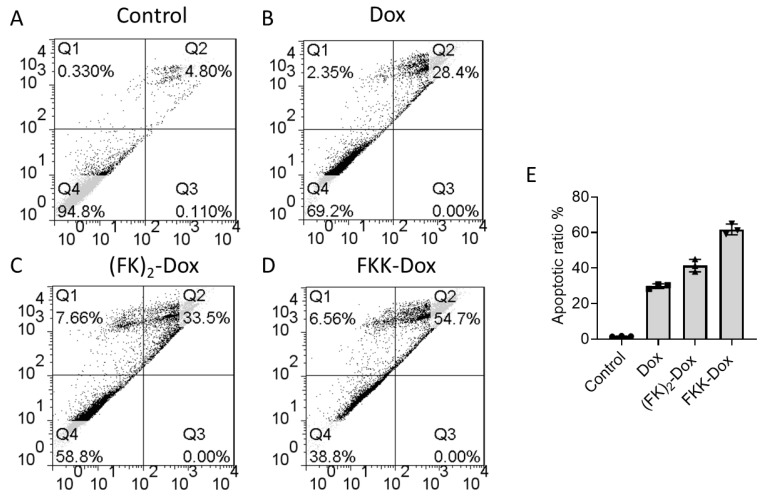
(**A**–**D**) Induction of apoptosis by different kinds of Dox species in Hela cells. Hela cells were treated with free-Dox (100 nM), (FK)_2_-Dox (100 nM of Dox) or FKK-Dox and analyzed 24 h after the treatment. Raw data for a representative experiment was shown. (**E**) The apoptotic cell ratio in different groups of cells after treated with free-Dox (100 nM), (FK)_2_-Dox (100 nM of Dox) or FKK-Dox. Three independent replications were performed. Error bars are SD.

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
