# Peer review of "Small Peptide-Doxorubicin Co-Assembly for Synergistic Cancer Therapy"

_molecules, 2020, doi:10.3390/molecules25030484_

Round 1

Reviewer 1 Report

Li et al. reported the preparation and characterization of small-peptide Doxorubicin co-assemblies, their characterization and their evaluation in terms of cell internalization and cytoxicity.
The paper is well conceived and the results well presented, therefore I reccomend the publication on Molecules, upon some minor revisions.
In particular, the draft should be polished by an English native speaker (e.g. for tenses, mispellings etc…).
Page 3 line 106: in which conditions the authors obtained the reported PDI and size? Did they use DLS? pease specify
Page 3 line 106: (KF)2 stands for what? Pease specify
Page 4 line 123: (FK)2 stands for what? Is it the same of (KF)2?
Page 6: in vitro cytotoxicity studies. Toxicity studies should be done also on KF and KFF alone to assess that the observed effects are not due to the positive charge of the peptide.
Page 9: preparation of co-assemblies. Please indicate the used concentrations
Page 9: In vitro dox release: Please add the concentrations that were used.

Reviewer 2 Report

The work of Yanga et al. concerns small peptide-Doxorubicin co-assembly in order to decrease its (Dox) toxicity and increase its therapeutic efficacy.

After reading the main text I have some doubts and suggestions:  

1. The anticancer activity of (FK)2-dox and FKK-dox against HeLa, a cervical cancer cell lineA549, a lung cancer cell line, and MDA-231, a breast cancer cell line, was evaluated. Authors concluded that " the cellular uptake of Dox is enhanced due to the positive charges of nanoparticles and that the release of Dox is pH-responsive, only in malignant cancer cells, the co-assembled nanoparticles de-assemble fast because of the acidic pH in the cancer cells, while in normal cells, despite the uptake occurs, the de-assembly kinetics is very slow"  I am wondering:

A) how Authors know that uptake occurs in normal cells , but the de-assembly kinetics is very slow as they only tested cancer cells ???

and B) cancer cells are known to acidify their environment, therefore the interior of cells themselves is alkalised. the question is if their the co-assembled nanoparticles (FK)2-dox and FKK-dox will de-assemble in tummor vessel before cellular uptake?Please comment it in the light of the final conclusions

2. The additional experiment with quantitative ROS measurements in cells undergoing oxidative stress after treatment
with (FK)2-dox and FKK-dox could be performed.

Reviewer 3 Report

Review of MS # molecules-683490

This manuscript addresses the co-assembled nanoparticles consisted Doxorubicin (Dox) with Fmoc-FK (FK) and Fmoc-FKK (FKK) are able to improve the therapeutic efficacy and mitigate the side effects of Doxorubicin. The authors indicated that these nanoparticles had homogeneous size distribution and positive charges after adjusting the component ratio of peptide and Dox, so making them efficiently uptake by tumor cells. They also showed that these nanoparticles are stable in alkaline and neutral conditions however de-assembly at acidic pH, enabling them selective Dox release in malignant cancer cells.

They also indicated that these nanoparticles induced cancer cell killing effects, apoptosis and cell cycle arrest, and they reduced side effects of Dox to normal cells. Please revise a minor point.

#1. Page 2, line 56. It had better to delete ‘the’.#2. Page 2, line 72. Is it better to use ‘chemotherapy drug’, not to use ‘chemotherapy’?

#2. Page 6, line 165. Please indicate p-values for Fig. 6A-C.

#3. For Materials and Methods, please indicate detailed information of cell lines, according to https://www.mdpi.com/journal/molecules/instructions.

Round 2

Reviewer 2 Report

The Authors of  manuscript entitled "Small peptide-Doxorubicin co-assembly for synergistic cancer therapy"  gave response to my comments as well as preformed additional experiment with ROS measurements. Although the Authors did not performed the experiment with normal cells as reference, I think  that they explanation and  rephrase of the final conclusions  are sufficient.

The paper is well conceived and the results well presented, therefore I reccomend the publication on Molecules in present form.